# Development of Vacuum-Steam Combination Heating System for Pasteurization of Sprout Barley Powder

**DOI:** 10.3390/foods11213425

**Published:** 2022-10-28

**Authors:** Seon Ho Hwang, Sung Yong Joe, Jun-Hwi So, Seung Hyun Lee

**Affiliations:** 1Department of Smart Agriculture Systems, Chungnam National University, Daejeon 34134, Korea; 2Department of Biosystems Machinery Engineering, Chungnam National University, Daejeon 34134, Korea

**Keywords:** sprout barley powder, vacuum-thermal combination, water activity, discrete element modeling

## Abstract

The processing of sprout vegetables in powder form has been known to extend the shelf-life by retaining nutritional values; however, sprout powder products were exposed to a variety of contaminants, such as microbial contaminants, during processing and storage. Therefore, the proper treatment for removing the contaminants in the powder was required without compromising their quality properties. This study was conducted (1) to determine a suitable pasteurization method for sprout barley powder, and (2) to investigate the effect of vacuum-steam heating combination treatment on the quality change and the lethality of microorganisms in sprout barley powder. The heating pattern of sprout barley powder was elucidated with a vacuum-thermal combination system consisting of a vacuum chamber, overhead stirrer, far-infrared heater, and PID (Proportional-Integral-Differential) controller. In addition, the mixing patterns of sprout barley powder, depending on the types of stirring blades, were evaluated by discrete element modeling using EDEM™ software. The vacuum-steam combination heating system was fabricated using the investigated pre-design factors. The quality change in sprout barely powder was evaluated by measuring the microbial inactivation, CIE values (*L**, *a**, *b**, Δ*E*), and water activity (a_w_). During the pasteurization process, steam could be directly injected into the chamber at regular intervals for two hours to transfer moisture and heat to the powder. By combining steam and vacuum conditions, the population of *E. coli* O157:H7 in the powder was reduced by 4.33 log CFU/g, eliminating all *E. coli* O157:H7 in the powder. In addition, the water activity (a_w_) of the powder was significantly decreased in a vacuum pressure environment without the quality deterioration.

## 1. Introduction

Recently, sprout vegetables have received great attention from consumers as health functional foods because they are rich in phytonutrients such as flavonoids, polyphenols, glucosinolates, isothiocyanates, proteins, minerals, and vitamins [1]. It was reported that the regular consumption of sprout vegetables could prevent inflammatory bowel and neurodegenerative diseases [2]. Sprout vegetables have been processed into various types of food such as rice cakes, additives in instant food, and powders. Among these products, the demand for sprout powder products with improved convenience of intake have been increased. 

However, the food powder products made from agricultural products are easily exposed to foreign material and microbial contamination during drying, pulverization, and storage processes [3,4]. In particular, a number of significant foodborne pathogens such as Bacillus cereus and *Escherichia coli* O157:H7, and foreign materials such as metallic substances generated during the pulverization process, have been detected in the sprout powder products [5]. Additional treatments for removing foodborne pathogens and foreign materials are required in the final stage of sprout powder processing; however, the inactivation of microorganisms in sprout powder is difficult due to the low moisture content in sprout powder and the increased heat resistance of microorganisms [6]. The excessive thermal treatment for inactivating microorganisms contained in the powder compromises its intrinsic organoleptic and physical properties [7]. Therefore, the optimal condition for the thermal treatment process should be determined to prevent quality deterioration in the powder products [8].

Steam is widely utilized as a heating medium in a variety of food processes owing to the high heat transfer coefficient and re-use of the latent heat of evaporation [9]. In food processing, steam treatment has remarkable effects, such as the inactivation of microbial load, the change in physical and mechanical properties, and the decrease in the activity of enzymes during storage. 

Vacuum treatment can minimize oxidation and browning of foods by removing moisture in the absence of oxygen [10]. Vacuums can create an inflated structure of food by expanding the air and water vapor present in the food [11] This structure can make it easy to evaporate moisture inside food. Since water can be evaporated at a low temperature under vacuum, vacuum treatment is suitable for the processing of heat sensitive food [12]. The rapid moisture evaporation inside food during vacuum treatment can prevent the growth and reproduction of microorganisms and can delay water-mediated degradation reactions [13]. The effect of vacuum-steam combination treatment on the inactivation of microorganisms in food products has been evaluated by several researchers [14,15,16,17]. The vacuum treatment could evaporate additional moisture into food products by steam injection, and the thermal energy provided by steam injection and moisture evaporation was able to inhibit the growth of microorganisms without food quality deterioration. However, the amount of powder samples used in the studies was not sufficient to apply the vacuum-steam combination treatment in the commercial food industry.

Since the batch thermal treatment of food powder products is significantly affected by the amount and moisture content of powders, it is necessary to heat the powder more uniformly. The non-uniform temperature distribution in the powder can cause the survival of microorganisms related to food safety concerns [18,19,20]. Powder mixing is one of the important operations to improve product homogeneity in the food industry, and the rotation effect can ensure the temperature uniformity of the powder [21,22]. The mixing patterns of food powder products can be affected by their physical properties (i.e., cohesive force, surface energy, density, and diameter); thus, an appropriate mixing blade should be selected, depending on the types of food powder products in question [23]. Discrete element method (DEM) simulation can possibly deduce the design factors of mixing blades by performing powder mixing prediction depending on the shape of the blade mixer and the physical properties of powders [24]. DEM powder mixing simulation can provide the mixing pattern between the shapes of blades and the powder. In addition, the assessment of powder mixing behavior by DEM can be applied to enhance the temperature uniformity of the powder during simultaneous mixing and heating processes.

This study was conducted to develop a vacuum-steam combination heating system that can pasteurize large-capacity powder, and to investigate the optimal condition of vacuum-steam combination heating for removing *E. coli* O157:H7 in sprout barley powder. Additionally, a preliminary experiment to identify the main factors of powder pasteurization and discrete element method simulation were performed to predict mixing patterns of powder, depending on the types of stirring blades.

## 2. Materials and Methods

### 2.1. Investigation of Design Factors for the Development of Vacuum-Heating System

#### 2.1.1. Sample Preparation for Pre-Experiment

Prior to the development of the vacuum-steam combination heating system for the pasteurization of sprout barley powder, preliminary experiments were performed to investigate the system design factors, and the microbial inactivation effect of the simultaneous vacuum-heating, using lab-scale vacuum heating system. The sprout barley powder produced by a domestic farm (Geongangjungsim Co., Ltd., Gimcheon, Korea) was purchased from the online market. The sprout barley powder of 100 g was used for each vacuum-heating experiment to determine changes in water activity and color.

In order to investigate the pasteurization effect of simultaneous vacuum-heating, the aliquot of *Escherichia Coli* O157:H7 (ATCC 35150) of 10 mL (initial concentration of approximately 9 log CFU/mL) was inoculated into sprout barley powder of 100 g inside a sterile filter bag (Filtra Bag, 17.78 cm × 30.48 cm, Labplas, St. Julie, QC, Canada). The inoculated powder samples were uniformly mixed by hand massaging for 30 min. The samples were transferred to a sterilized stainless-steel tray and then dried in a clean bench for 30 min at room temperature. The final cell concentration and water activity (aw) of the inoculated samples (control sample in the pre-experiment) were 8.0 to 8.1 log CFU/g and 0.336, respectively.

#### 2.1.2. Powder Pasteurization Method Using Experimental Scale Vacuum Heating System

The lab-scale vacuum heating system is shown in Figure 1a. The system consisted of a vacuum chamber and a pump (vacuum degassing chamber with pump, Labfirst Scientific Instruments, Laguna Hills, CA, USA), a stainless steel (SUS 304) heating chamber, an overhead stirrer (MS5060, Misung Scientific, Seoul, Korea), a far-infrared heater (Far infrared heater, MISUMI, Seoul, Korea), and a PID (proportional integral derivative) controller (PX9, Hanyoungnux, Incheon, Korea). The vacuum chamber could withstand vacuum pressure of up to 0.1 bar. The vacuum stirring seal installed in the center of the vacuum chamber could effectively prevent the inflow of air during heating. The heating chamber was installed at the bottom center of the vacuum chamber using two spacers. The distance between the heating chamber and the heater was 1 cm. In order to mix sprout barley powder, the I-shaped stirring bar was connected to an overhead stirrer through the vacuum stirring seal. The heating temperature was controlled using a PID controller connected to the heater, and two K-type thermocouples were installed to measure the temperatures of the sample and the heating chamber, as shown in Figure 1b. One of thermocouples was placed at a distance of 1 cm away from the outside bottom of heating chamber and one was placed in the inside heating chamber to measure the sample temperature.

The sprout barley powder samples were thermally treated under two different pressure conditions (101.325 kPa (≈ 1 bar) and 0.2 bar). The target heating temperatures were 60 °C, 80 °C, and 100 °C. The time required for the powder samples to reach 60 °C, 80 °C, and 100 °C at 1 bar and 0.2 bar were 45 s, 62 s, and 80 s, respectively. In the case of 0.2 bar, the times were 52 s, 86 s, and 91 s, respectively. The powder samples were heated for 15, 30, and 60 min after reaching the target heating temperatures. Uniform temperature distribution in the sample was obtained by rotating the stirring bar at a speed of 100 rpm. The sample amount for each experiment was 100 g. The inoculated powder samples were also treated with the same experimental protocol. As soon as the experiment was completed, the powder sample of 1.5 g was put in a sterilized petri dish to measure water activity (a_w_) and color. In addition, the inoculated powder sample of 100 g was immediately transferred into a sterile stomacher bag filled with 900 mL of 0.2% peptone water. After uniform mixing and appropriate series dilution, the diluent of 1 mL was spread onto *E. coli*/Coliform dry film medium (coliform count plate/petrifilm™ *E. coli*, Aerobic, 3M™, St. Paul, MN, USA). Subsequently, the film medium was incubated at 37 °C for 24 h, and bacterial numbers were counted.

### 2.2. Discrete Element Method Simulation

A Discrete Element Method (DEM) simulation was carried out to predict the mixing pattern of powder particles in the chamber depending on the types of stirring blades. Since DEM simulation has been widely exploited to predict the behavior of multiple solid particles interacting with mechanical components, such as chamber walls, blades, and impellers, it is suitable for modeling the mixing pattern of small particle materials, such as food powder, in the mixing chamber [25]. In this study, the mixing powder sample was the most important process so that the powder sample could be uniformly heated. EDEM software (EDEMTM 7.1, Altair Engineering, Inc., Troy, MI, USA) was utilized to simulate the mixing patterns of the powder depending on the types of stirring blades, finally selecting the optimal stirring blade for uniform mixing of the powder.

#### 2.2.1. Conditions of DEM Simulation Parameters

In order to perform the mixing simulation of sprout barley powder, the Generic EDEM Material Model (GEMM) Database of EDEM was employed to select the physical properties and shape of the powder particles. The particle size distribution of the sprout barely powder was measured using the dry sieve method. The measured particle size of the powder using the sieve shaking method was approximately 200 mesh (corresponding to 0.074 mm). However, the diameter of the particles applied to the simulation was set to 0.1 mm to reduce simulation analysis time (Figure 2). The total particle number applied to the simulation was 100,000 due to the size of the chamber geometry. The main parameters of the simulation, such as particle density, static friction, and particle-geometry factors, are listed in Table 1.

The original size of the chamber for mixing and heating sprout barely powder was 50 cm in height and 40 cm and 30 cm in the top and the bottom diameter, respectively. The chamber geometry used in the simulation was reduced to a scale of approximately one-twenty seventh of the original size, as shown in Figure 3. Two different colors (dark cyan and light yellow) particles were sequentially filled in the chamber geometry to observe the mixing patterns of particles, depending on the types of stirrer blades. The custom-designed four types of stirrer blades used in the simulation are displayed in Figure 4. The rotation speed of the blade in the simulation was 100 rpm—the same as the actual speed used in the experiment. The rotation speed was set by taking into account the stirring blade, the weight of the powder, and the additional moisture. The stirrer manufactured to fit the size of the chamber was stirred at a speed of 100 rpm inside the chamber with respect to the central axis. Figure 4a,b are the modified paddle blade and the screw-ribbon combined blade (ribbon blade installed perpendicular to the wall). The scrub paddle-shaped blade capable of scraping the walls of the chamber is shown in Figure 4c. Figure 4d is the screw conveyor blade combined with the ribbon blade to enable the horizontal and vertical transfer of powder.

#### 2.2.2. Contact Model of the Simulation

DEM is the practical method to understand the behavior of individual particles in mixing processing. Newton’s second law of motion was applied to the DEM simulation to evaluate the powder mixing in the vacuum-steam combination heating system. The translational and rotational motions (i.e., velocity and acceleration) of individual powder particles were governed by the following equations:(1)$$m_{i}\frac{dv_{i}}{dt}=\overset{n}{\underset{j}{\sum }}\left(F_{ij}^{n}+F_{ij}^{t}\right)+m_{i}g$$
(2)$$I_{i}\frac{d\omega_{i}}{dt}=\overset{n}{\underset{j}{\sum }}\left(R_{i}\times F_{ij}^{t}-\tau_{ij}^{r}\right)$$
where $R_{i}$, $m_{i}$, $v_{i}$, and $\omega_{i}$ are the radius (m), mass (kg), translational speed (m/s), and rotational speed (m/s) of the particle, respectively. $F_{ij}^{n}$ and $F_{ij}^{t}$ are vertical and horizontal contact forces due to particle-to-particle or particle-wall collisions of the Hertz-Mindlin model implemented in EDEM [26]. $\tau_{ij}^{r}$ is the torque owing to the rolling frictional resistance [27].

Hertz’s equation was applied to determine the normal force vector due to impact [26]:(3)$$F_{n}=\frac{4}{3}E^{*}\sqrt{R^{*}}\delta^{3/2}$$
where $E^{*}$ is the effective Young’s modulus $\left(\frac{1}{E^{*}}=\frac{1-v_{1}{}^{2}}{E_{1}}+\frac{1-v_{2}{}^{2}}{E_{2}}\right).$ $E_{1}$, $E_{2}$, $v_{1}$, and $v_{2}$ are contacts in the individual Young’s modulus (Pa) and Poisson’s ratio (dimensionless) of the individual particles. $R^{*}$ is the effective radius (m), defined as $\frac{1}{R^{*}}=\frac{1}{1/R_{1}}+\frac{1}{1/R_{2}}$. $R_{1}$ and $R_{2}$ are the radii (m) of the individual contact particles. δ is normal overlap vector (m).

The tangential force vector due to impact was estimated by Mindlin’s equation as follows:(4)$$F_{t}=-8G^{*}\sqrt{R^{*}\delta_{n}}\delta_{t}$$
where $G^{*}$ is the equivalent shear modulus (Pa), $R^{*}$ is the effective radius (m), $\delta_{n}$ is the normal overlap vector(m), and $\delta_{t}$ is tangential overlap vector (m).

Rolling friction can be defined as the particle’s resistance to the torque produced by particles on the contact surface [28] Thus, the torque was estimated by using the rolling friction coefficient ($\mu_{r}$):(5)$$\tau_{\iota }=\mu_{r}F_{n}R_{i}\omega_{\iota }$$
where $\tau_{\iota }$ is the torque (N∙m) acting on the particle, $\mu_{r}$ is the rolling friction coefficient, $F_{n}$ is the normal contact force (N), $R_{i}$ is the radius (m) of the particle, and $\omega_{\iota }$ is the angular velocity (rad/s).

The normal damping force of a particle was determined by the following equations [26]:(6)$$\beta =\frac{\ln e}{\sqrt{ln^{2}e+\pi^{2}}}$$
(7)$$S_{n}=2E^{*}\sqrt{R^{*}\delta_{n}}$$
(8)$$\vec{F_{n}^{d\;}}=-2\sqrt{\frac{5}{6}}\beta \sqrt{S_{n}m^{*}}\vec{v_{n}^{rel}}$$
where e is the coefficient of restitution, $E^{*}$ is the equivalent Young’s modulus ($N/mm^{2}$), $R^{*}$ is the effective radius (m), $\delta_{n}$ is the normal overlap, m is the equivalent mass (kg), and $v_{n}^{rel}$ is the relative vertical velocity (m/s).

The Hertz-Mindlin model, combined with the JKR model, was employed to determine a normal force affecting the agglomeration between particles due to the size (especially fine size) and water content of powder particles [26].
(9)$$F_{JKR}=-4\sqrt{\pi \gamma E^{*}}\alpha^{\frac{3}{2}}+\frac{4E^{*}}{3R^{*}}\alpha^{3}$$
where γ is the surface energy and α is the particle contact area.

### 2.3. Vacuum-Steam Combination Heating System

Based on the results from the preliminary study using the experimental scale vacuum heating system, the vacuum-steam combination heating system was designed and fabricated to uniformly pasteurize sprout barley powder, as shown in Figure 5 and Figure 6. The heating system consisted of a vacuum chamber and pump, vertical motor, stirring blade (fabricated based on the result of DEM simulation), band heater, and a water jacket heater. The truncated cone-shaped vacuum chamber (400 mm in top diameter, 300 mm in bottom diameter, and 500 mm in height, stainless steel (SUS304)) was fabricated to withstand a vacuum pressure of up to 0.1 bar. The oil-less type vacuum pump was able to maintain a vacuum pressure range between 86.66 and 99.99 kPa (≒ 0.1 bar) in a space between 80 L and 90 L. The vertical motor was suitable to rotate the powder at 100 rpm with a sufficient output of 5 HP (3.7 kW). A screw conveyor blade, combined with the ribbon blade (Figure 4d), was manufactured for mixing the powder. In addition, the blade was made of stainless steel (SUS 304). The steam injection was controlled by a solenoid valve and relay and was programmed so that steam of 0.6 mL could be injected into the powder for 5 s. The steam pressure and temperature were maintained at 5 bar and 150 °C, respectively. The band heater and water jacket heater were installed on the outside wall of the chamber. The water temperature was effectively increased using the band heater, and the heated water played an important role to minimize the environmental factor to affect the change in the inside temperature of the chamber. In addition, it was possible to maintain the uniform heating temperature with an accuracy of 0.01 °C using a precise PID controller and heat exchanger.

### 2.4. Foodborne Pathogen Inoculum Preparation

A certain strain of *Escherichia Coli* O157:H7 (ATCC 35150) was used in this experiment. The strain was obtained from the department of food science and technology culture collection at Chungnam National University (Daejeon, South Korea). *E. coli* O157:H7 was mixed with 50% glycerol and Luria-Bertani (LB) medium in a ratio of 1:1. The mixture was stored in a stock (*v*/*v*) form in a cryogenic freezer at −80 °C. The 50% glycerol stock (*v*/*v*) was plated on LB agar to obtain a single colony of *E. coli* O157:H7. After plating, the strain was incubated at 37 °C for 24 to 48 h, and then stored in a refrigerator at 4 °C. For all experiments, the single colony of *E. coli* O157:H7 was inoculated into LB medium of 250 mL and incubated with shaking at 200 rpm and 37 °C for 24 h to obtain a suitable form for inoculation into powder. Finally, the concentration of *E. coli* O157:H7 in the inoculum solution was measured to be 8 to 9 log CFU/g.

### 2.5. Sprout Barley Powder Sample Preparation and Inoculation

The sprout barley powder was purchased from a domestic sprout barley powder processing company (Aenong Co., Ltd., Jinan, Korea). The powder was pulverized into a size of 200 mesh (corresponding to 0.074 mm) in an air mill machine. The initial water activity (aw) of the powder was approximately 0.235. The powder was packaged in a 10 kg bulk bag and stored at room temperature (around 21 to 23 °C). Two different powder samples (inoculated with/without *E. coli* O157:H7) were used in this study. The initial coliform population of the uninoculated sample of 100 g was 4.33 ± 0.238 log CFU/g. To inoculate *E. coli* O157:H7 into the powder sample, the powder of 1 kg was filled in the sterile plastic bag (Biohazard Bags, 30 cm × 45 cm, Korea Ace Scientific, Seoul, Korea) containing *E. coli* O157:H7 inoculum solution of 100 mL. The inoculated powder sample was properly mixed by massaging for 30 min. And then the sample was put into the vacuum-steam combination heating chamber, and was stirred at atmospheric pressure for 1 h prior to the experiment. The initial average population of *E. coli* O157:H7 in the inoculated powder sample was 5.38 log CFU/g. The inoculated powder sample of a total 5 kg was used in the experiment.

### 2.6. Vacuum-Steam Combination Heating Treatment

The effectiveness of the developed combination heating treatment on microbial inactivation in the powder sample of 5 kg was investigated under different treatment conditions (Figure 7). In order to uniformly inoculate *E. coli* O157:H7 into the 5 kg powder, the inoculated powder sample was stirred at atmospheric pressure and room temperature for 1 h prior to the experiment. The vacuum pressure and heating temperature were maintained at 0.2 bar and 85 °C, respectively. In order to obtain uniform heating temperature, the temperature sensor of PID was installed inside the band heater. The total heating time was 2 h after the heating temperature reached 85 °C. The stirring blade was rotated at 100 rpm. The different amounts of steam (0, 60 mL, and 120 mL) were injected into the chamber filled with the powder sample in a heating-vacuum state (note that the volume of steam injection was the amount of the steam condensed into water). The steam of 60 mL and 120 mL was injected at 25 and 10 min intervals, respectively. At the interval, a five-time application of 5 s on/off steam injection was programmed using a relay controller. After the final steam injection, the powder samples were treated by vacuum drying for 20 and 40 min, respectively.

### 2.7. Microbial Enumeration

The powder samples (100 g) treated with different conditions were immediately collected after all the experiments. The powder sample was placed in a sterile filter bag (Filtra Bag, 17.78 cm × 30.48 cm, Labplas, St. Julie, QC, Canada) filled with sterilized buffered peptone water of 600 mL and then homogenized in a stomacher blender for 3 min. After homogenizing, the 1 mL aliquot of the sample was serially diluted in 9 mL of 0.1% Phosphate buffered saline, and the diluent of 1 mL, were spread on a dry *E. coli*/Coliform film (coliform count plate/petrifilm™ *E. coli*, Aerobic, 3M™, St. Paul, MN, USA). Dry films were incubated at 37 °C for 24 h, and the *E. coli* population was enumerated by counting red and blue colonies.

### 2.8. Water Activity and Color Measurements

Changes in water activity (aw) and the color of sprout barley powder were investigated depending on different treatment conditions. Water activity of the powder was measured using a water activity meter (AquaLab PRE, METER Group Inc., Pullman, WA, USA). CIE values of the powder were analyzed using a colorimeter (NR60CP Colorimeter, 3nh., Shenzhen, China). The total color change (∆*E**) was calculated by the following equation:(10)$$\Delta E^{*}={\left[{\left(L^{*}-L_{0}^{*}\right)}^{2}+{\left(a^{*}-a_{0}^{*}\right)}^{2}+{\left(b^{*}-b_{0}^{*}\right)}^{2}\right]}^{1/2}$$
where *L**, *a**, and *b** express whiteness/darkness, redness/greenness, and yellowness/blueness, respectively. The subject “0” indicates the CIE values of the untreated powder samples.

### 2.9. Statistical Analysis

The Statistical Software SPSS Version 26 (SPSS Inc., Chicago, IL, USA) was utilized to carry out statistical computations for the inactivation of *E. coli* O157:H7, the changes in water activity, and color. Duncan’s multiple range tests were employed to determine the significant differences between the control sprout barley powder sample and the treated powder samples by different heating conditions at *p* < 0.05.

## 3. Results and Discussion

### 3.1. Comparison of Vacuum and Atmospheric Pressure Heat Treatment Using Experimental Scale System

#### 3.1.1. Changes in Temperature and Color of Powder

Temperature profiles of sprout barley powder treated by different heating temperatures (60 °C, 80 °C and 100 °C) and pressures (1 bar and 0.2 bar) are shown in Figure 8. The time required for the powder samples treated at 0.2 bar to reach all target temperatures were slightly longer compared to 1 bar due to changes in pressure and volume of air in the chamber; however, after reaching target temperatures, the target temperatures were constantly maintained regardless of pressure conditions. The color values of sprout barley powder before/after the treatments are summarized in Table 2. As a result of measuring the color of the CIE values of the powder (control sample) before treatment, *L**, *a**, and *b** were 57.61 ± 2.57, −3.78 ± 0.62, and 26.39 ± 2.68, respectively. Regardless of pressure conditions, the *L** value (lightness) of the powder sample showed a similar brightness to the initial brightness over time at 60 °C and 80 °C, but significantly decreased at 100 °C. The *a**(redness/greenness) values of the powder samples treated at 0.2 bar, different heating temperatures, and times were slightly different from the value of the untreated powder samples; however, the value of the powder sample treated at 1 bar and 100 °C was increased, with an increase in heating time, due to browning. On the other hand, the *b** values (yellowness/blueness) of the powder samples treated at 0.2 bar were similar to the value of untreated samples, regardless of heating temperatures and time. The *b** values of the powder samples treated at atmospheric pressure and 100 °C were significantly increased compared to the value of the untreated sample. While the moisture of the powder was quickly evaporated at high temperature and low pressure (vacuum pressure), magnesium in the chlorophyll of sprout barley powder could be replaced with hydrogen, thereby resulting in the conversion of chlorophylls to pheophytins [29]. The range of the total color change (∆*E**) values of the treated powder samples by different conditions was between 2.09 and 6.74. The ∆*E** value showed the tendency to increase as heating temperature increased. Figure 9 shows the pictures of the powder samples before/after the treatments. The discoloration of the powder samples could not be distinguished with the naked eye. The commercial value of sprout barley powder has been mainly decided by the color (especially greenness). Therefore, the high temperature (100 °C), which could lead to browning or scorching of the powder, was not suitable for the heating treatment of the powder, regardless of the pressure condition.

#### 3.1.2. Changes in Microorganism Population and Water Activity of Sprout Barley Powder

Figure 10 shows the reduction of *E. coli* O157:H7 population in the sprout barley powder samples treated by different heating conditions. *E. coli* O157:H7 inoculum solution of 10 mL was inoculated into sprout barley powder of 100 g. After inoculation, the concentration of *E. coli* and the water activity in the powder sample were approximately 8.1 ± 0.301 log CFU/g and 0.336, respectively.

The populations of *E. coli* in the powder samples treated at 60 °C and 1 bar for 15, 30, and 60 min were reduced by 1.54 ± 0.07, 2.17 ± 0.186, and 1.89 ± 0.379 log CFU/g, respectively. The log reductions of *E. coli* in the powder samples treated at 60 °C and 0.2 bar were 1.63 ± 0.09, 1.10 ± 0.06, and 1.87 ± 0.365 log CFU/g, respectively. The inactivation effect of pressure with an increase in treatment time was not significant at 60 °C.

The log reductions of *E. coli* in the powder samples treated at 80 °C and 1 bar for 15, 30, and 60 min were 1.67 ± 0.121, 2.59 ± 0.117, and 2.64 ± 0.126, respectively. The log reductions obtained from 30 and 60 min were not significantly different. At 80 °C and 0.2 bar treatment condition, 1.93 ± 0.09, 2.18 ± 0.07, and 2.57 ± 0.133 log CFU/g were inactivated for 15, 30, and 60 min, respectively. Similar to the results from 80 °C and 1 bar conditions when treated up to 60 min, the heating times didn’t affect the inactivation of *E. coli.*

In the case of 100 °C, 1 bar, and 0.2 bar treatment conditions, the log reduction values of *E. coli* in the powder were mostly higher than other treatment conditions. The heating temperature has a significant effect on the inactivation of *E. coli* O157:H7; however, except for the treatment temperature of 100 °C, no significant change in microbial inactivation was observed when the treatment time reached up to 60 min.

Figure 11 shows the changes in water activity of the powder samples treated by different heating conditions. After thermal treatment, the lowest and highest water activities in the powder samples were 0.098 and 0.259, respectively. The changes in water activity in the treated powder samples were highly dependent on the treatment temperature; however, the lower the temperature, the more they were affected by the pressure. Since the moisture in the powder samples, which played an important role as the heat transfer medium, evaporated rapidly in a vacuum (low pressure)-thermal environment, the significant inactivation effect on *E. coli* in the powder samples treated at 0.2 bar was not observed. The thermal resistance of microorganisms in dried powder was much higher than that of ordinary powder products [6], and water activity in food is an important indicator that determines the growth of microorganisms during storage or distribution [3,4,30]. However, the removal of a large amount of water content in the powder could result in the fine degradation of the powder particles and an increase in the amount of lipids exposed to air. The excessive thermal treatment of low-moisture powder products or dried powder products could affect the quality deterioration by causing oxidation and promoting the browning of the powder products [9]. Therefore, it is important to maintain an appropriate water activity during the powder pasteurization process. Based on the results from preliminary experiments using the lab scale system, the steam injection part was equipped with the vacuum–heating system. In addition, the low pressure (0.2 bar) and a heating temperature range between 80 °C and 90 °C were determined as the optimal pasteurization conditions for sprout barley powder in order to ensure microbial safety with minimizing quality deterioration.

### 3.2. Powder Mixing Patterns Depending on the Types of the Stirring Blade

The mixing patterns of sprout barley powder in the vacuum heating chamber, depending on the types of stirring blades, are shown in Figure 12. The selection of the proper stirring blade type was important to uniformly heat the powder without caking of the powder. The simulated powder mixing pattern using the paddle blade is shown in Figure 12a. Since the paddle blade occupied the central space of the chamber, the powder particles were stacked at the edges of the chamber, and overall mixing of the powder particles was not observed. The stacking of the powder particles could cause caking of powder particles and non-uniform heat transfer between the particles. Therefore, the paddle blade was not suitable for application for the thermal treatment of the powder samples. In the case of the screw-ribbon blade, a screw for vertical reversal of the powder particles located in the center of the chamber and a circular-helical blade for moving the powder particles to the wall side of the chamber were combined to compensate for the disadvantages of the paddle (Figure 12b). The rotation of the screw-ribbon blade led the powder particles, located the bottom of the chamber, to diffuse upwards. However, from the side view, the powder mixing occurred only in the space occupied by the screw-ribbon blade. Figure 12c shows the simulated powder mixing pattern using the scrub-paddle blade. Although the design of the scrub-paddle blade was able to smoothly flip the position of the powder particles and mix to the edges of the chamber, the unbalanced shape of the stirrer rotated at high speed could cause eccentricity. The stacking of particles at a certain area in the chamber could occur due to this eccentricity. Figure 12d shows the powder mixing pattern using the screw conveyor blade, considering decantation, according to the shape of the chamber. The two screw blades were adapted to the chamber so that the powder particles were properly mixed. In addition, the blades installed vertically from the chamber wall could increase the contact area between the particles and the inner surface of the chamber. The heat transfer and mixing of the powder particles could be maximized from the heating source (the surface of the chamber). In addition, the screw conveyor could produce effective powder mixing regardless of the shape of the powder particles and could exhibit better mixing behavior as the amount of powder increased [31,32]. Based on the results obtained from the powder mixing simulation, the screw conveyor that could enhance heat transfer between the powder particles and the heating source with uniform mixing was fabricated to be applied to the vacuum-steam heating system.

### 3.3. Inactivation Rate

Figure 13 shows the inactivation rate of *E. coli* O157:H7 in sprout barley powder samples depending on the steam injection volume (0, 60 and 120 mL). The initial *E. coli* O157:H7 concentration of inoculated sprout barley powder was 5.38 log CFU/g. The coliform concentration of sprout barley powder was 4.33 log CFU/g. As shown in Figure 13a, *E. coli* O157:H7 in the powder samples thermally treated with/without steam injection were totally inactivated. By inoculating *E. coli* O157:H7 solution to the powder samples, the large amount of inoculum solution affected the increase in water activity and could improve heat transfer between the heating surface of chamber (inner chamber wall) and the powder sample (Figure 14). Therefore, it was feasible to inactivate all *E. coli* O157:H7 in the powder samples. In order to investigate the coliform inactivation effect of the vacuum-steam combination heating system, the powder samples were treated with/without steam injection. The log reduction of coliform in the powder sample treated without steam was only 0.70 log CFU/g (Figure 13b). As vacuum heating progressed, the moisture content in the powder sample, which is the heat transfer medium, was rapidly evaporated. Therefore, the vacuum heating had no effect on the microbial inactivation of powder samples. The log reduction of coliform in the powder samples treated with steam injections of 60 and 120 mL were 2.01 log CFU/g and 4.33 log CFU/g, respectively. As the pasteurization process progressed, the water vapor evaporated from the powder was consistently increased and became saturated inside the chamber. Saturation could occur when the ratio of the water vapor evaporated from the powder and the ejected air amount (by vacuum) in the chamber became equal. After reaching saturation point, the ejected air amount was increased. Since steam was supplied when the surface water vapor pressure of the powder began to decrease, the saturation point was retained in the chamber. Therefore, there was not a decrease in the surface water vapor pressure of the powder, and the water activity of the powder was maintained similar to the state of the initial powder or slightly reduced. The steam injection could transfer high thermal energy to the powder sample, and could maintain the water activity of the powder, which could be used as a heat transfer medium. The thermal energy and moisture supplied by the steam injection were effective at removing microbial contaminants in the powder samples.

Figure 14 shows the change in water activity of the powder before and after thermal treatment. The initial state was the water activity conditions of the untreated powder samples. The inoculated state was the water activity conditions of the inoculated powder samples. The treatment state was the powder subjected to each vacuum-steam heating. The initial water activity of the powder samples was measured to be about 0.235 and 0.564 at the initial status and after inoculation, respectively. Figure 14a shows the change in water activity of inoculated samples at different steam injection volumes (0 and 60 mL). After the steam injection treatment, the water activity of the powder without steam injection (0 mL) was 0.204, and the powder injected with steam (60 mL) was 0.569. The similar level of microbial inactivation was achieved in the population of *E. coli* by both treatment conditions. It seems that the water activity of the sample was increased due to the additional moisture in the inoculum that provided sufficient pasteurization even without steam. In Figure 14b, the water activity values of the un-inoculated samples after steam injection (0, 60 and 120 mL) were measured to be 0.132, 0.213, and 0.347, respectively. While the large amount of steam injection (120 mL) could inactive all *E. coli* in the powder, it also resulted in an increase in water activity compared to before treatment. However, after steam pasteurization, the powder could have a safe water activity value through sufficient vacuum drying time. Figure 15 shows the water activity (a_w_) change of the sprout barley powder without steam injection as vacuum heating time increased. The a_w_ of the powder sample was significantly reduced from 0.25 to 0.15 to up to 30 min, and then slightly decreased. In the vacuum heating environment, the a_w_ of 5 kg of sprout barley powder could be reduced to 0.131 at the most, which meant that there was a very small amount of moisture in the powder sample. After 30 min of vacuum heating, the moisture content in the powder was lowered, finally decreasing heat transfer for deactivating the microorganism. The experimental results indicated that steam injection significantly affected microbial inactivation in the powder products containing low moisture content. In addition, the vacuum-heating was able to strongly evaporate the remaining moisture, finally improving the quality of the powder. Therefore, it could be suggested that the application of vacuum and steam techniques was proven effective to pasteurize *E. coli* O157:H7 in large-capacity powder products.

### 3.4. Color Change Assessment of Treated Powder

Table 3 shows the CIE values of sprout barley powder treated by each experimental condition. The initial values of the chromatic parameters of the sprout barley powder used in the experiment were $L^{*}$ = 50.93 ± 3.26, $a^{*}$= −7.51 ± 1.75, and $b^{*}$ = 32.66 ± 2.16. The inoculated powder heated without steam spray had *L**, *a**, and *b** values of 47.95, −5.94, and 32.47, respectively, and the total change ∆*E* was 3.37. The *L**, *a**, and *b** values of the inoculated powder treated with 60 mL of steam were changed to 39.79, −3.63, and 31.13, respectively, and the total change value (∆*E*) was 11.89. The effect of steam injection on the color change of the powder was found to be greater than that of artificial water injection (inoculation solution). The values of *L*, a**, and *b** of the un-inoculated powder that was heat-treated without steam injection were determined to be 50.41, −8.12, and 32.47, respectively, and the ∆*E* value was 4.94. The values of *L** and *a** did not show a significant difference from the existing ones, and the value of *b** showed a significant increase in yellowness due to high temperature at vacuum pressure. The color changes of the powder injected with 60 mL of steam showed that the values of *L**, *a**, and *b** were 4.387, −2.48, and 37.27. For the powder injected with 120 mL of steam, 44.28, 2.09, and 36.03 were measured, respectively. The total color change values (∆*E*) were 9.62 and 12.16 at 60 and 120 mL, respectively. The original color of the powder (closer to green) appeared to be darker due to the significantly reduced lightness (*L**). Figure 16 shows the visual appearance of the powder before and after vacuum-steam treatment. Although the CIE values of powder showed a significant change, the discoloration of the powder sample could not be discerned visually. Therefore, it could be suggested that using a vacuum steam heat pasteurizer was appropriate for securing the quality of powder products.

## 4. Conclusions

In this study, the pasteurization effect of the developed vacuum-steam combination heating system on *E. coli* O157:H7 and coliform in the sprout powder was evaluated. Results of DEM simulations have confirmed that the screw conveyor ribbon type could effectively mix a large number of small particles.

Additionally, *E. coli* O157:H7 in the inoculated sample was completely inactivated, demonstrating the powder mixing and steam effects of the developed agitator. The water activity was investigated as an important pasteurization factor to inactivate *E. coli* O157:H7 in the sprout barley powder. Appropriate steam injection could significantly reduce the population of *E. coli* O157:H7 by providing rehydration and additional bactericidal effect to the powder. In particular, the steam injection of 120 mL reduced *E. coli* O157:H7 in 5 kg of powder by more than 4.33 log CFU/g. In addition, heating in an environment of low pressure quickly evaporated moisture in the powder and prevented deterioration of product quality. The developed vacuum-steam combination heating system could improve the pasteurization effect and improve the drying effect of powdered food.

## Figures and Tables

**Figure 1 foods-11-03425-f001:**
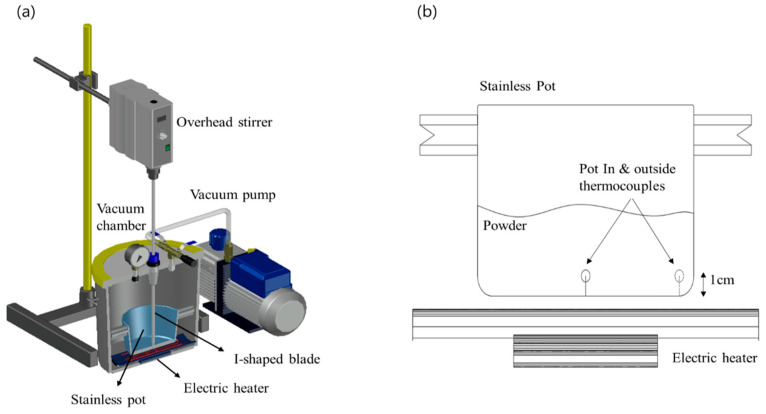
Configuration and appearance of the vacuum heating system (**a**): schematic diagram of the system and (**b**) the internal layout of the vacuum chamber.

**Figure 2 foods-11-03425-f002:**
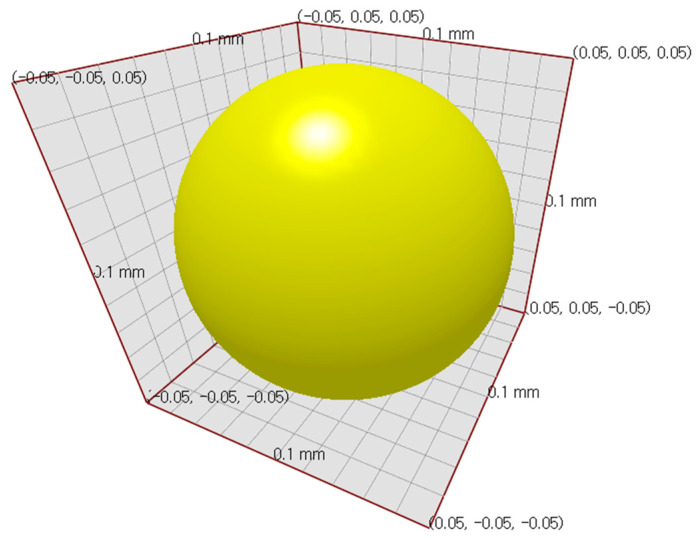
Particle shape used in the DEM simulation.

**Figure 3 foods-11-03425-f003:**
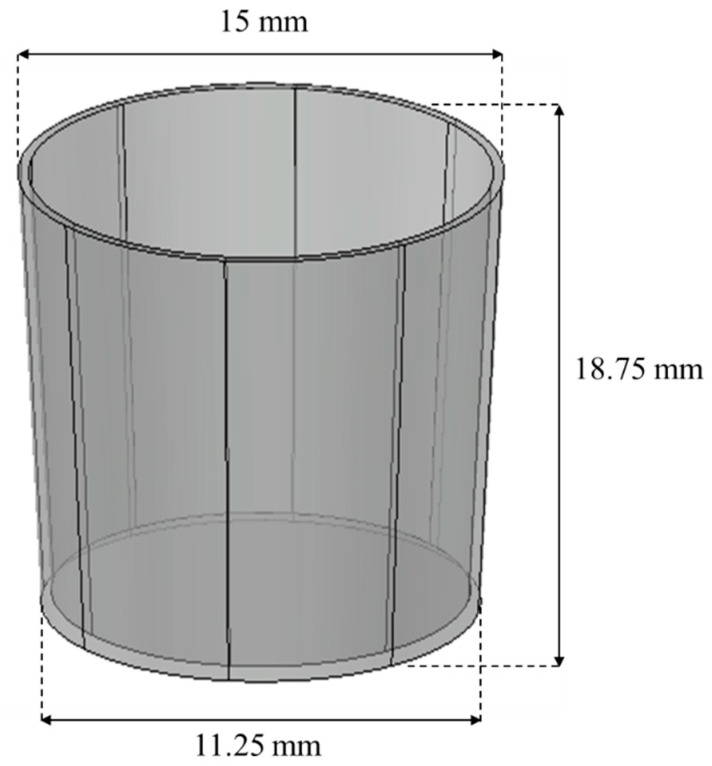
The dimension of chamber geometry.

**Figure 4 foods-11-03425-f004:**
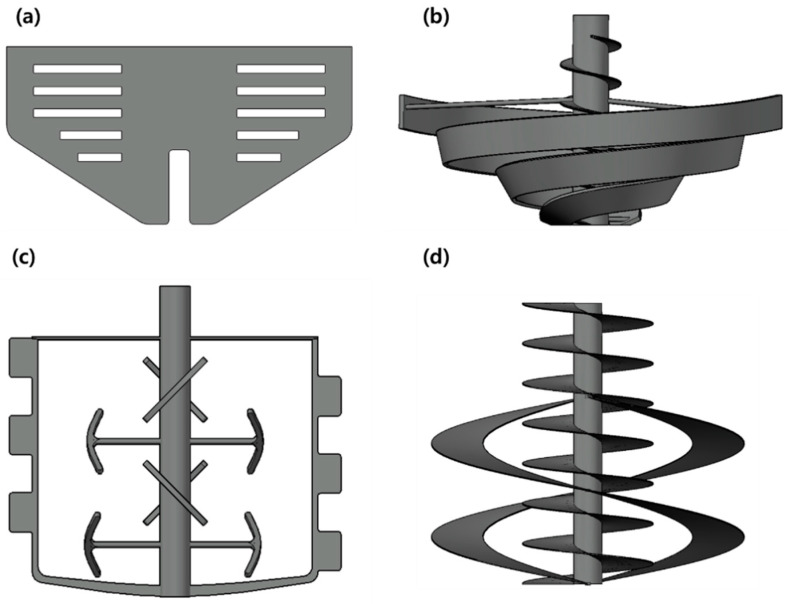
Types of stirring blades used in the simulation: (**a**) modified paddle, (**b**) screw-ribbon, (**c**) scrub paddle-shaped, and (**d**) screw conveyor.

**Figure 5 foods-11-03425-f005:**
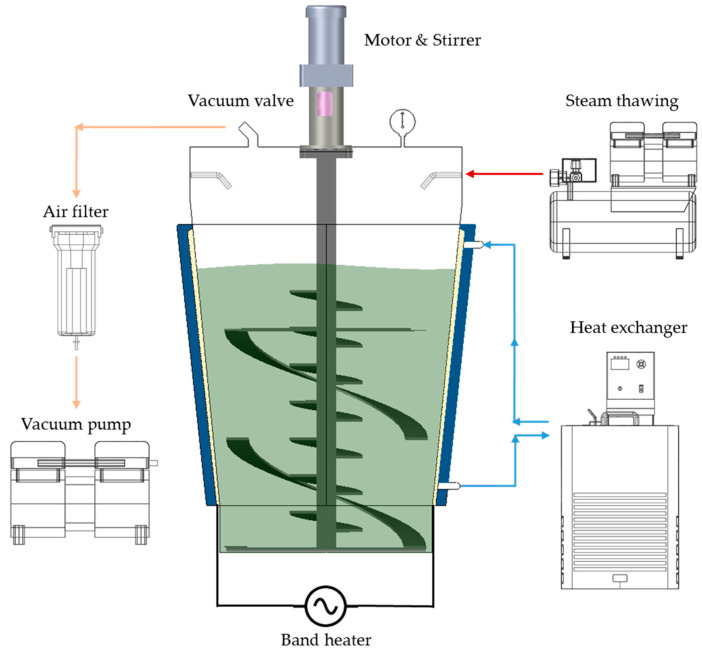
A schematic diagram of the vacuum-steam combination heating system.

**Figure 6 foods-11-03425-f006:**
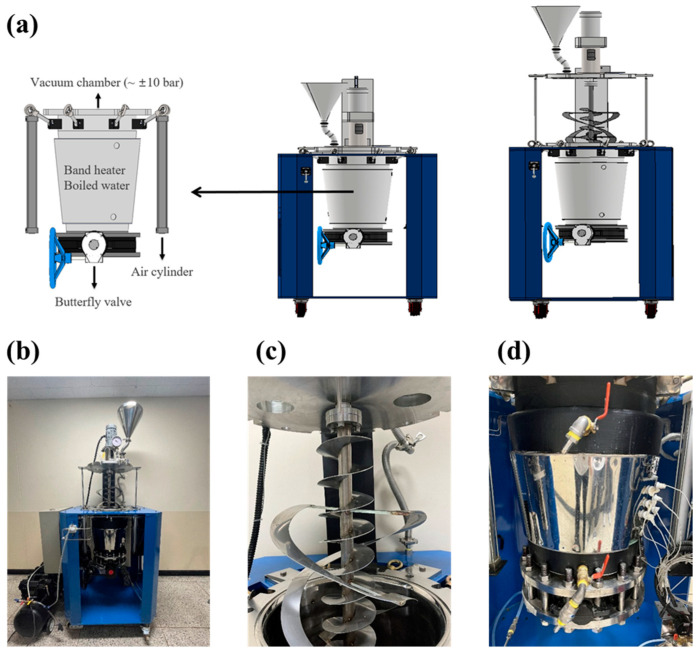
A schematic diagram of the developed vacuum-steam heating system: (**a**) a 3D schematic, (**b**) a front view, (**c**) a conical ribbon blade, and (**d**) a vacuum chamber.

**Figure 7 foods-11-03425-f007:**
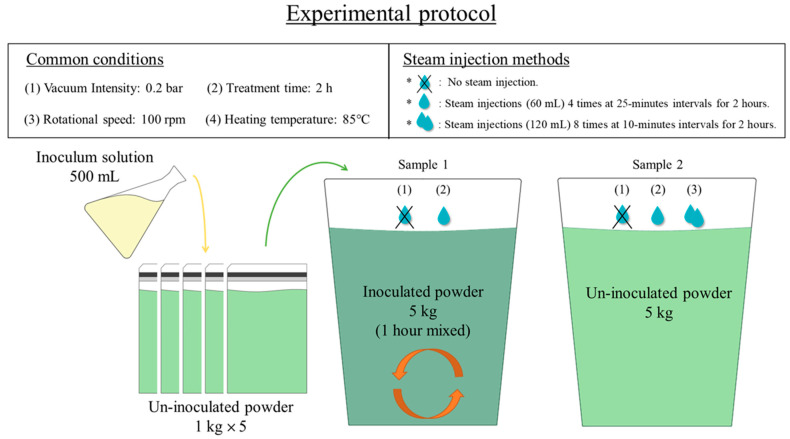
Schematic diagram of the experimental protocol.

**Figure 8 foods-11-03425-f008:**
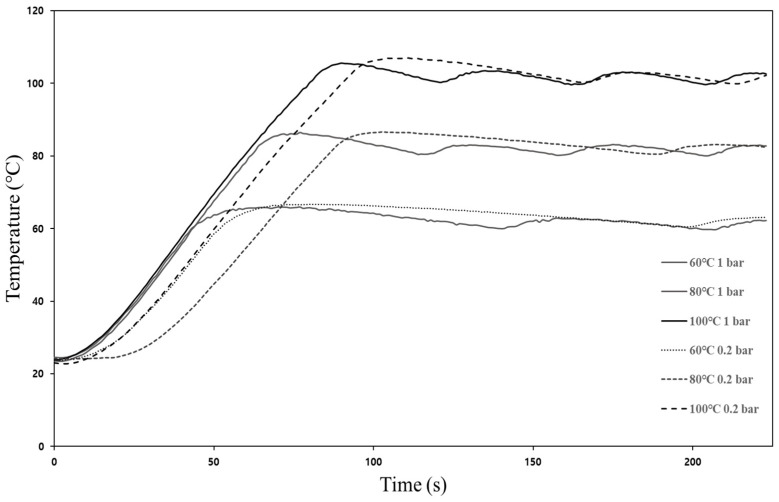
The temperature profiles of sprout barley powder samples treated by different heating conditions.

**Figure 9 foods-11-03425-f009:**
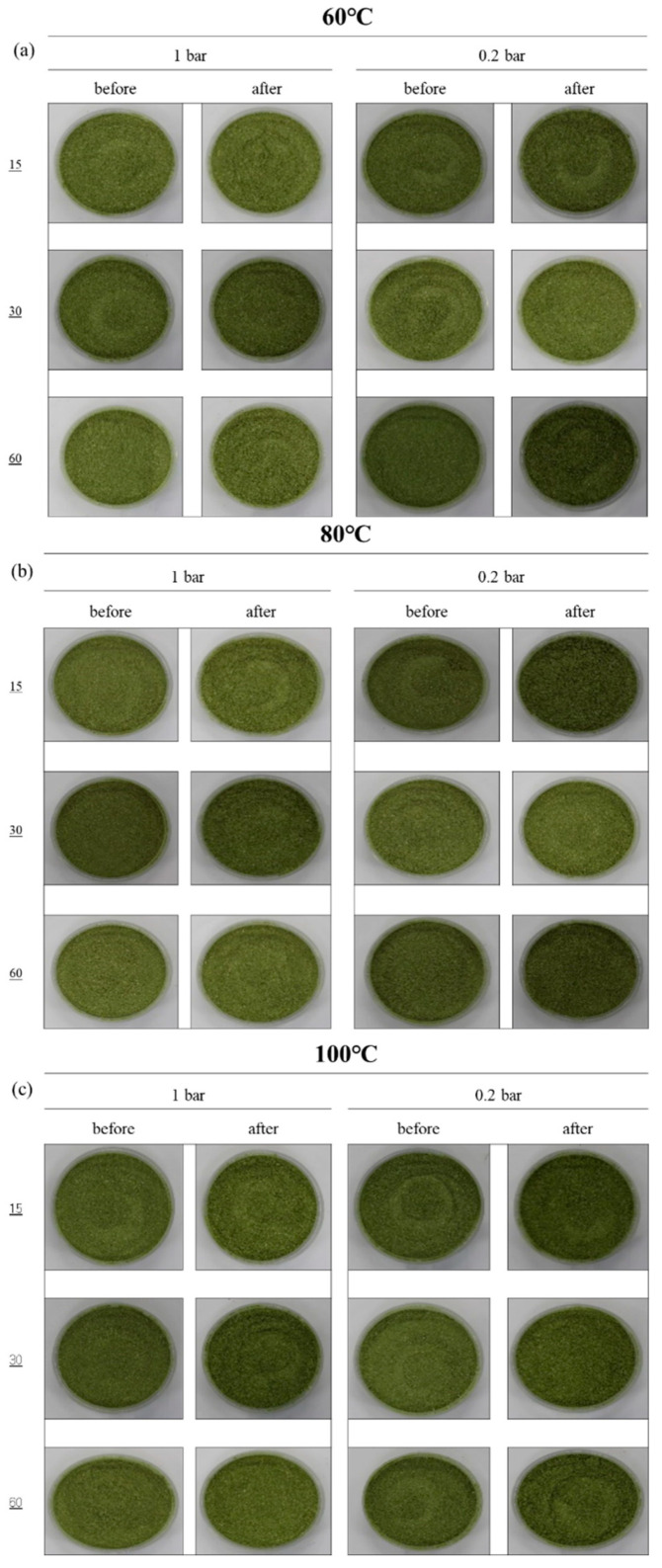
The images of sprout barley powder samples before and after treatments with different heating conditions.

**Figure 10 foods-11-03425-f010:**
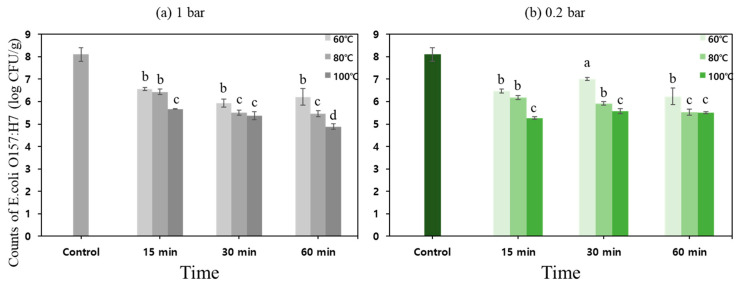
The inactivation of *E. coli* O157: H7 in sprout barely powder treated by different heating condition: (**a**) 1 bar, (**b**) 0.2 bar. Different letters indicate significant differences (*p* < 0.05).

**Figure 11 foods-11-03425-f011:**
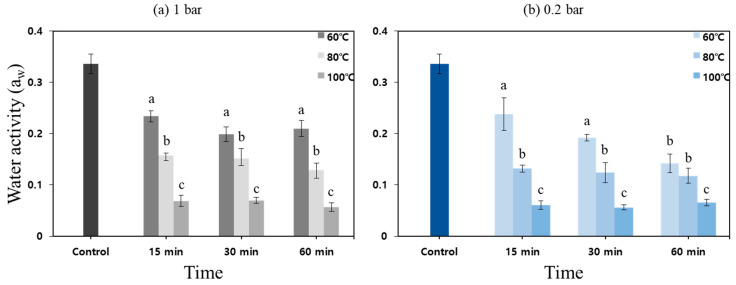
The change in water activity of the sprout barley powder by pressure ((**a**) 1 bar, (**b**) 0.2 bar), treatment time (15, 30 and 60 min), and heating temperature (60, 80 and 100 °C). Different letters indicate significant differences (*p* < 0.05).

**Figure 12 foods-11-03425-f012:**
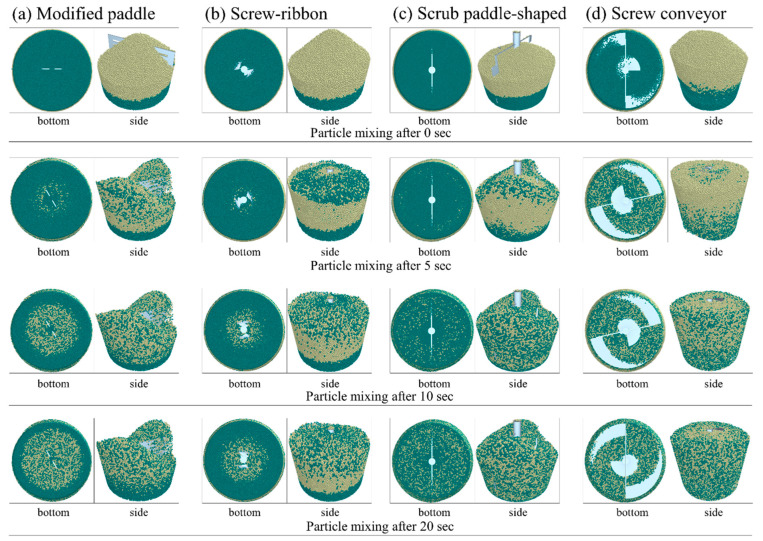
Simulated mixing patterns of powder particles at 0, 5, 10, and 20 sec: (**a**) modified paddle, (**b**) screw-ribbon, (**c**) scrub paddle-shaped, and (**d**) screw conveyor.

**Figure 13 foods-11-03425-f013:**
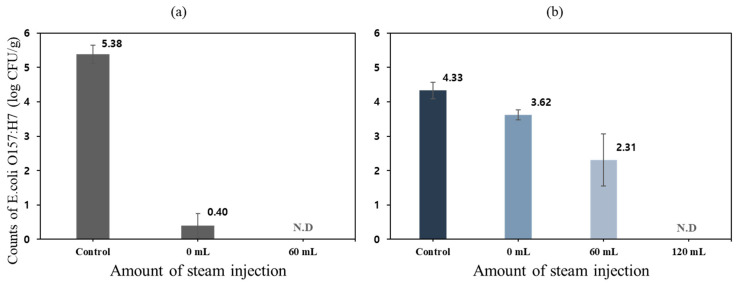
The inactivation rate of microbial contaminants in barely powder samples: (**a**) *E. coli* O157:H7 and (**b**) coliform.

**Figure 14 foods-11-03425-f014:**
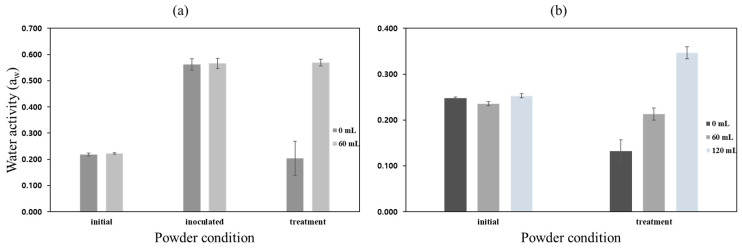
The change of water activity of the (**a**) inoculated and (**b**) un-inoculated powder by vacuum-steam treatments (0, 60, and 120 mL).

**Figure 15 foods-11-03425-f015:**
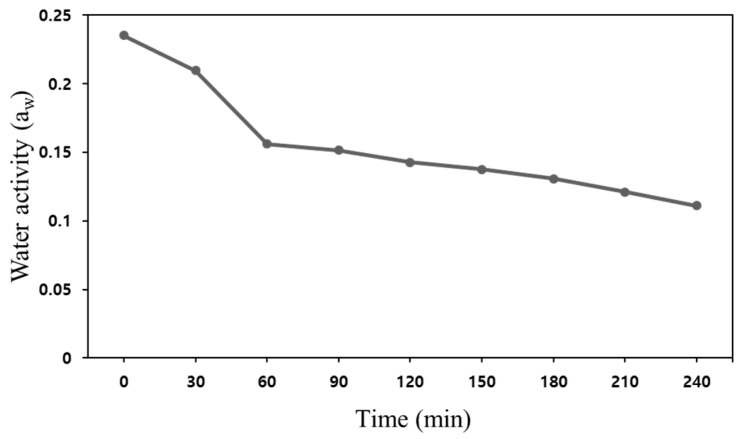
Change in water activity at vacuum heating time (0 to 240 min).

**Figure 16 foods-11-03425-f016:**
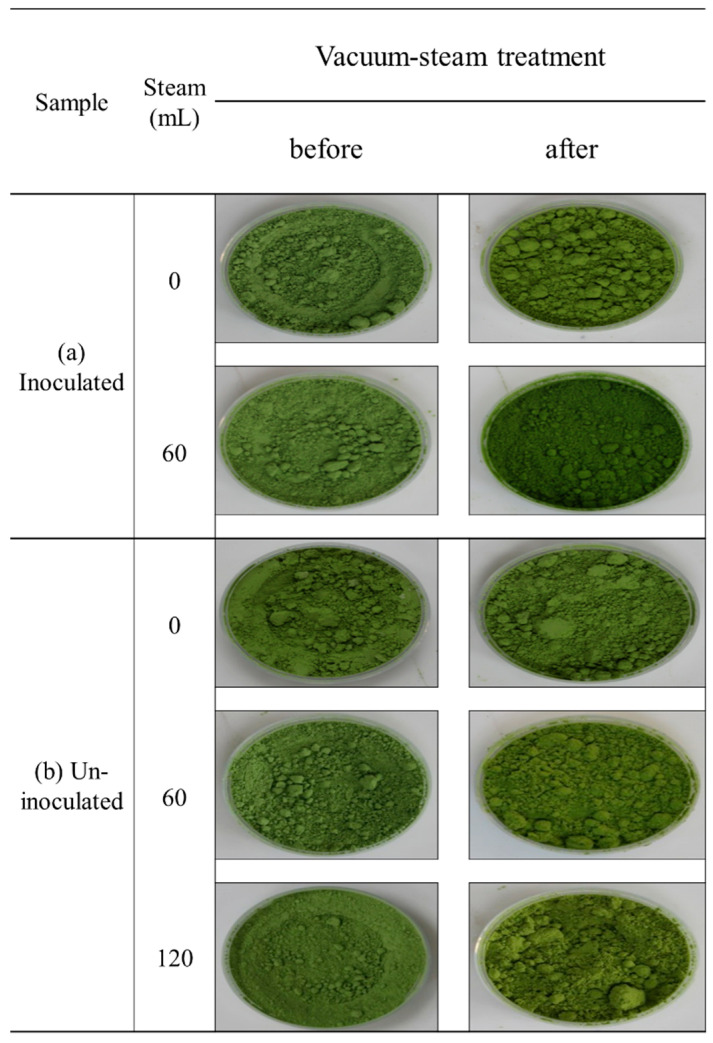
Conditions before and after treatment of (**a**) inoculated and (**b**) un-inoculated sprout barley powder as a function of by vacuum-steam treatment (0, 60, and 120 mL).

**Table 1 foods-11-03425-t001:** Parameters used in the DEM simulation.

Parameters	Value
Total particle number	≤100,000
Mixer operating speed (rpm)	100
Particle diameter (mm)	0.1
Particle density (kg/m^3^)	1500
Particle shear modulus (Pa)	10^7^
Particle Poisson’s ratio	0.25
Geometry density (kg/m^3^)	4000
Geometry shear modulus (Pa)	10^10^
Geometry Poisson’s ratio	0.25
Particle-particle static friction coefficient	0.32
Particle-particle rolling friction coefficient	0.15
Particle-particle restitution coefficient	0.75
Particle-Geometry static friction coefficient	0.45
Particle-Geometry rolling friction coefficient	0.15
Particle-Geometry restitution coefficient	0.5

**Table 2 foods-11-03425-t002:** Changes in CIE values and total color difference of sprout barley powder treated by different heating conditions.

Control (Initial Values)			*L**	*a**	*b**	*ΔE*
Time	Pressure	Temperature	57.61 ± 2.57 ^abc^	−3.78 ± 0.62 ^abc^	26.39 ± 2.68 ^ab^	-
15 min	1 bar	60 °C	58.46 ± 3.29 ^ab^	−3.46 ± 0.29 ^bc^	25.59 ± 1.92 ^ab^	3.49 ± 1.31 ^ab^
		80 °C	57.85 ± 3.46 ^ab^	−3.29 ± 0.73 ^cd^	24.45 ± 1.98 ^a^	4.07 ± 1.16 ^b^
		100 °C	52.85 ± 0.73 ^de^	−2.80 ± 0.42 ^d^	26.2 ± 0.60 ^ab^	4.11 ± 0.69 ^b^
	0.2 bar	60 °C	58.90 ± 1.68 ^ab^	−3.60 ± 0.86 ^a^	25.93 ± 2.91 ^ab^	3.29 ± 0.79 ^ab^
		80 °C	54.70 ± 1.05 ^cd^	−3.24 ± 0.22 ^cd^	26.88 ± 3.58 ^ab^	4.44 ± 1.05 ^b^
		100 °C	53.30 ± 2.01 ^de^	−3.76 ± 0.50 ^abc^	30.65 ± 0.76 ^c^	5.03 ± 2.69 ^c^
30 min	1 bar	60 °C	58.45 ± 2.32 ^ab^	−3.95 ± 0.26 ^ab^	26.59 ± 1.2 ^ab^	2.48 ± 0.68 ^a^
		80 °C	59.61 ± 1.55 ^a^	−3.82 ± 0.39 ^abc^	25.46 ± 1.58 ^ab^	2.50 ± 1.83 ^a^
		100 °C	51.39 ± 1.72 ^f^	−1.81 ± 0.17 ^e^	26.51 ± 0.22 ^ab^	6.74 ± 1.44 ^e^
	0.2 bar	60 °C	57.15 ± 1.40 ^abc^	−3.27 ± 0.59 ^cd^	25.6 ± 2.60 ^ab^	2.90 ± 0.51 ^a^
		80 °C	55.15 ± 2.16 ^bcd^	−3.18 ± 0.28 ^cd^	27.25 ± 3.53 ^ab^	2.45 ± 0.39 ^a^
		100 °C	52.31 ± 1.20 ^ef^	−3.75 ±0.57 ^abc^	31.01 ± 0.54 ^c^	4.93 ± 2.86 ^c^
60 min	1 bar	60 °C	59.78 ± 2.01 ^a^	−4.14 ± 0.38 ^a^	27.18 ± 0.65 ^ab^	2.82 ± 1.28 ^a^
		80 °C	56.65 ± 1.34 ^bcd^	−3.40 ± 0.68 ^bc^	25.23 ± 0.75 ^a^	2.09 ± 0.68 ^a^
		100 °C	52.88 ± 2.41 ^de^	−1.67 ± 0.36 ^e^	26.88 ± 0.57 ^ab^	5.97 ± 1.29 ^d^
	0.2 bar	60 °C	57.26 ± 0.33 ^abc^	−3.60 ± 0.77 ^abc^	25.72 ± 2.93 ^ab^	2.80 ± 0.79 ^a^
		80 °C	59.05 ± 3.29 ^ab^	−3.52 ± 0.32 ^abc^	28.09 ± 3.75 ^b^	3.44 ± 1.48 ^ab^
		100 °C	52.9 ± 1.03 ^de^	−3.48 ± 0.56 ^abc^	31.13 ± 0.97 ^c^	5.08 ± 2.45 ^c^

^a–f^ Means in the same column with different letters are significantly different at *p* < 0.05.

**Table 3 foods-11-03425-t003:** Changes in CIE values and total color difference in (a) inoculated and (b) un-inoculated powder by vacuum-steam treatment (0, 60, and 120 mL).

Control (Initial Values)		*L**	*a**	*b**	Δ*E*
Sample	Steam (mL)	50.93 ± 3.26 ^a^	−7.51 ± 1.75 ^a^	32.66 ± 2.16 ^a^	-
(a) Inoculated	0	47.95 ± 0.03 ^a^	−5.94 ± 0.02 ^b^	32.47 ± 0.01 ^a^	3.37 ^a^
	60	39.79 ± 2.03 ^c^	−3.63 ± 0.89 ^c^	31.13 ± 1.39 ^a^	11.89 ^d^
(b) Un-inoculated	0	50.41 ± 2.71 ^a^	−8.12 ± 1.21 ^a^	37.53 ± 0.53 ^b^	4.94 ^b^
	60	43.87 ±1.36 ^b^	−2.48 ± 0.27 ^d^	37.27 ± 1.21 ^b^	9.62 ^c^
	120	44.28 ± 1.67 ^b^	−2.16 ± 0.06 ^d^	36.03 ± 0.48 ^b^	9.17 ^c^

^a–d^ Means in the same column with different letters are significantly different at *p* < 0.05.

## Data Availability

Data presented in this study are available in the article.

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
