# Peer review of "Development of Vacuum-Steam Combination Heating System for Pasteurization of Sprout Barley Powder"

_foods, 2022, doi:10.3390/foods11213425_

Round 1
Reviewer 1 Report
The authors use the term pasteurization / sterilization interchangeably, which is incorrect and should be clearly specified in the paper. For example, in the title of the work, we have the word "pasteurization" and at the end of the introduction, "sterilization" (L86). When can we talk about sterilization? And when about pasteurization? This should be clarified.
Research methodology inconsistent with the presented research results.
Abstract.
The abstract should concern information resulting directly from the conducted research. The authors did not investigate the nutritional value of the powder either before or after treatment. Research has not been conducted in this direction.
L25, L42-43 and everywhere. Latin names of pathogens are written in italics.
L18. PID? Explain the abbreviation for the first time using it in the text of the article.
L25-28. The authors give a specific value for the reduction of E. coli, but from what starting level?
Introduction
L37-39. What products are they? Write specifically.
L86-90. The text should be rewritten. What was the purpose of the research?
Materials and Methods
L93-107. What was the purpose of posting the results of this experiment? Was it a control sample? Under what conditions was the drying carried out (specify parameters)?
L108. "Sterilization"? In the title of manuscript Authors used “pasterization”?
L166. Figure 1 and L159. Simulation was set to 0.1 mm, isn’t it? The particle in Figure 1 has a rounded shape, so why are 2 dimensions inserted instead of the diameter? For what purpose was it decided to present this drawing?
L264-267. Did the addition of 100 mL inocilum solution cause surface dissolution of the powder?
L273-284 and Figure 5 (experimental protocol). The information contained in the text is not consistent with the figure. Why is no blending used in the case of un-inoculated powder? After all, it is known that mixing facilitates heat transport and moisture removal. Thus, “un-inoculated powder “ will be considerably less favorable. Was drying "(4) " used for both variants? Why was steam injection (120 mL) not used for inoculated powder?
L305-306 and everywhere. Remove the "^" from the color parameters.
L323-324. What was the control sample? What parameters? Should this be precisely specified in the methodology?
L346. Figure 8. Increase the font size for the legend data. We use unit abbreviations according to the SI system of measures. "Time (sec)" it should be "Time (s)".
L348. Table 2. Please modify the table so that the name "control" and the values ​​of the color parameters are in one line. What was the control sample?
L353. Figure 6. What was the sample "before"? Why are there as many as 18 sample variants "before"? Authors presented values ​​of color parameters in the table 2 for only 1 control sample?
L370-371. Unclear. In relation to what?
L361-377. Discussion is needed.
L380-381. How was the pressure affected? (See L403- Fig. 8). According to statistics, it is irrelevant.
L382-384. Unclear. On the basis of what data is this statement? Add a reference to the drawing.
L395-398. Temperature 90oC? The research concerned the temperature of 60, 80, 100oC?
L446-448. On the basis of what data is this statement? Please refer to the figure / figures. "... heating surface and the powder sample." Heating surface of what?
L487. Figure 11. Unclear names of samples. Where did this data come from? Explain what the names 'initial' and 'treatment' mean and where do so many variants come from?
L515. Table 3. Please modify the table so that the name "control" and color parameter values ​​are in one line. What was the control sample?
L520. Figure 13. What was the "before" sample, since one variant of the control sample was given for the color (Table 3). This has to be carefully explained in the methodology.
Professional linguistic proofreading recommended.
Author Response
The authors appreciate the reviewer’s constructive comments and suggestions. The manuscript was revised to incorporate the review comments. As suggested by the reviewer, the relevant sentences have been rewritten and have been presented in a different color (blue) in the revised manuscript. Please refer to the attached file.

Reviewer 2 Report
General remarks
The paper describes results of powder treatments by heat and steam including some simulations with DEM. However, the paper itself contains many repetitions regarding treatment conditions and others. On the other hand, some of the treatment conditions are poorly described. Results are sometimes confusing and need some more explanations. Especially the balances regarding added water and steam on one side and the effects on water activity of the powder needs clarification. For this purpose, I would strongly recommend to include the moisture contents of the powders at different states, e.g. before and after treatment, in the results section. Otherwise, the difference in water activity after the treatment with and without steam (figure 11, left hand side) cannot be explained. The increase from about 0.2 to 0.58 only by adding 60 ml of water to 5 kg of powder is not comprehensible. Furthermore, it has to be considered that most of added steam is removed by the vacuum pump.
Special remarks (not conclusive)
L 12: Why are metallic contaminants mentioned here? They cannot be removed by the treatment described. It should be removed.
L 23-24: The volumes of steam given here makes no sense. Is the volume related to the amount of water evaporated to steam or the volume of steam itself? In the latter case the steam mass depends on steam pressure and temperature. Assuming 100 °C and 1 bar, the mass of 60 ml of pure steam is only approx. 0.1 g, whereas approx. 60 ml of water gives 60 g. Anyway, the amount of steam added, preferably in g, should be related to the total amount of powder treated (5 kg).
L 41: References given here are not related to sprout powder.
L 89: Sterilization was not investigated in this study, because treatment conditions are far away from those required to sterilize the powder.
L 97: More information and specification of the sprout barley powder are necessary, e.g. moisture content, particle size distribution, and more.
L 101: Source of the E. Coli bacteria? Specification?
L 109-124: This paragraph contains mostly repetitions and it can be distinctly shortened.
L 126: Why here ‘atm’? Only SI units should be used.
L 126-128: How long was the time to reach the target temperature?
L 137-138: What was the minimum CFU per g, which could be determined with the method?
L 156-158: A particle size distribution cannot be measured by using only one sieve. After sieving using mesh 200, 2 different masses or portions are obtained, lower and higher than 0.074 mm. This is not a distribution. Nevertheless, not even the portions smaller than 0.074 mm are given by the authors. This is not sufficient to characterize the powder in any way.
L 164: What is the source of the parameters used in DEM?
L 166: This is the second Fig. 1 (see line 140). Why?
L 166: Why here a sphere with a diameter of 0.2 mm, but a diameter of 0.1 is mentioned in the text before? What about the third dimension? Why spheres here, although powder particles made from plant material often have shapes that are more fibrous?
L 175: The authors should explain why they used the 100 rpm for simulation. To obtain a more realistic simulation, at least the peripheral speed of the original stirrer, which is relevant for speeds and forces acting on powder particles, should be considered. At least the differences to the simulation approach have to be discussed.
L 230: See remark to L 126.
L 259: How was the powder pulverized?
L 260: Why here mesh 200 with 0.6 to 0.7 mm? Before, it was 0.074 mm exactly.
L 264: What were the standard deviations of the CFU counts?
L 279: How was the steam generated and what were the steam conditions, e.g. pressure and temperature after generation?
Fig. 5: Steam injection methods are not clear. E.g. 60 ml: Does it mean that 15 ml steam were injected after 25 min, 15 ml after 50 min, 15 ml after 75 min, and 15 ml after 100 min. Did it start at 0 min and end after 75 min? See also remark to lines 23-24 above.
L 325: This statement is not true, because considering treatment at 0.2 bar and 80 °C, there is an increase of L* during treatment from 54.7 (15 min) to 55.15 (30 min) up to 59.05 after 60 min.
L 348-349: Meaning of the small letters has to be explained in the legend. What is the first row of color data related to? Control? This has to be added. Considering that same letters in the same row are related to non-significant differences and vice versa, the information is not clear for L* and treatment time of 15 min. 54.70 +/- 1.05 is not significantly different from 58.46 +/- 3.29, because both have letter a. On the other hand, 54.70 +/- 1.05 should be significantly different from 53.30 +/- 2.01 with letter b. Considering the differences in the mean values, this is not plausible.
L 360: Why here different data to line 107.
L 361-377: These paragraphs can be distinctly shortened, since a full repetition of the data presented in the diagram is not necessary.
Fig. 9: As I understand the figure, the first line shows the states before start of the mixing device (0 seconds). Since the same geometry and the same procedure to generate and place the balls in the chamber are applied, the initial state before rotation should be the same. However, there are huge differences between a and d. Why?
L 441: The numbering of the figures does fit the figures in the paper, e.g. figure 13, but the highest figure is number #12.
L 441-449: The explanation for the higher inactivation rate in the bigger chamber compared to the pre-experiments makes no sense, because in both experiments the same ratio of powder and inoculation suspension of 10:1 was used.
Fig. 10: The effect of steam is not plausible, considering the ratios of powder, inoculation suspension and steam. 5 kg of powder in the chamber should contain 500 ml of suspension (mostly water) at the beginning. Considering that the 60 ml of steam are related to the initial water used for steaming, this is less than 15 % of the water added initially. Furthermore, it can be expected that the water added during the inoculation evaporates at the beginning of the treatment resulting in a lot of steam generation, which is much more than the steam added in the first periods of steam injection. Therefore, I don’t expect significant effects of the injection of 15 ml of steam (per time) into 5 kg of powder.
Author Response

(The authors gave the same response as above.)

Round 2
Reviewer 1 Report
Authors responded to the reviewer's comments and made appropriate corrections to the text of the paper as well as to figures and tables. I have no comments to manuscript.
Author Response
The authors appreciate the reviewer’s for thoughtful feedback and constructive suggestions. Reviewer’s comments have been very careful and useful for improving the manuscript.

Reviewer 2 Report
Although the authors tried to address most of the comments, the manuscript still contains serious flaws. The main one is related to the mass balance of water during treatment and the resulting water activity after treatment (figure 14 a). Firstly, the inoculation significantly increases the water activity due to the addition of 500 g water to 5 kg of powder. If no steam is added, water activity decreases to values below the initial ones due to drying at vacuum (0.2 bar 85 °C) as expectable. However, adding about only 60 g of steam during drying, where parts of water are removed (see sentence before), results in the same water activity as in the inoculated powder before drying. What is the source of the additional water? This is still not addressed by the authors. However, this is important, because some of the inactivation effects were explained by the differences in water activity.
Another relevant issue is the combination of absolute pressure (0.2 bar) and temperature (85 °C) for the vacuum drying. The evaporation temperature of water at 0.2 bar is about 60 °C. Therefore, the steam injected during vacuum drying cannot contribute to a significant increase of the powder’s water content, too.
Author Response
First of all, the authors appreciate the reviewer’s constructive comments. Please find and refer to the attached file.
